# Symptom-Dependent Changes in MEG-Derived Neuroelectric Brain Activity in Traumatic Brain Injury Patients with Chronic Symptoms

**DOI:** 10.3390/medsci9020020

**Published:** 2021-03-25

**Authors:** Don Krieger, Paul Shepard, Ryan Soose, Ava M. Puccio, Sue Beers, Walter Schneider, Anthony P. Kontos, Michael W. Collins, David O. Okonkwo

**Affiliations:** 1Department of Neurological Surgery, University of Pittsburgh, Pittsburgh, PA 15232, USA; PuccAM@UPMC.EDU (A.M.P.); okonkwodo@upmc.edu (D.O.O.); 2Department of Physics and Astronomy, University of Pittsburgh, Pittsburgh, PA 15232, USA; shepard@pitt.edu; 3Department of Otolaryngology, University of Pittsburgh, Pittsburgh, PA 15232, USA; soosrj@UPMC.EDU; 4Department of Psychiatry, University of Pittsburgh, Pittsburgh, PA 15232, USA; BeersSR@upmc.edu; 5Department of Psychology, University of Pittsburgh, Pittsburgh, PA 15232, USA; wws@pitt.edu; 6Department of Sports Medicine, University of Pittsburgh, Pittsburgh, PA 15232, USA; akontos@pitt.edu (A.P.K.); collinsmw@upmc.edu (M.W.C.)

**Keywords:** CamCAN, TEAM-TBI, post-concussion syndrome, normative atlas, test-retest reliability, insomnia, depression, anxiety, somatization

## Abstract

Neuroelectric measures derived from human magnetoencephalographic (MEG) recordings hold promise as aides to diagnosis and treatment monitoring and targeting for chronic sequelae of traumatic brain injury (TBI). This study tests novel MEG-derived regional brain measures of tonic neuroelectric activation for long-term test-retest reliability and sensitivity to symptoms. Resting state MEG recordings were obtained from a normative cohort, Cambridge Centre for Ageing and Neuroscience (CamCAN), baseline: *n* = 619; *mean* 16-month follow-up: *n* = 253) and a chronic symptomatic TBI cohort, Targeted Evaluation, Action and Monitoring of Traumatic Brain Injury (TEAM-TBI), baseline: *n* = 64; *mean* 6-month follow-up: *n* = 39). For the CamCAN cohort, MEG-derived neuroelectric measures showed good long-term test-retest reliability for most of the 103 automatically identified stereotypic regions. The TEAM-TBI cohort was screened for depression, somatization, and anxiety with the Brief Symptom Inventory and for insomnia with the Insomnia Severity Index. Linear classifiers constructed from the 103 regional measures from each TEAM-TBI cohort member distinguished those with and without each symptom, with *p* < 0.01 for each—i.e., the tonic regional neuroelectric measures of activation are sensitive to the presence/absence of these symptoms. The novel regional MEG-derived neuroelectric measures obtained and tested in this study demonstrate the necessary and sufficient properties to be clinically useful—i.e., good test-retest reliability, sensitivity to symptoms in each individual, and obtainable using automatic processing without human judgement or intervention.

## 1. Introduction

Traumatic brain injury (TBI) is a major cause of death and disability. Localized neuroelectric correlates of persistent functional sequelae after TBI would provide significant clinical value for diagnosis, disease monitoring, and targeted therapy.

It has been the informed expectation for a century that the keys to understanding the human brain will be found in measuring and understanding the electrical activity of neurons. Today, clinical neurophysiologists routinely measure single neurons to aid implantation of therapeutic devices deep in the brain [1]. Epileptologists use arrays of implanted “stereo EEG” electrodes and the population recordings obtained from them to diagnose and guide the treatment of intractable seizure disorders [2].

Population neuronal activity is presumed to be the basis for human behavior. Stereo EEG and comparable invasive methods produce voltage recordings with resolution of a few millimeters at best from up to a few hundred recording sites. Because the electric field interacts strongly with the conducting tissue in the brain, these measures are difficult to localize for currents at a distance from the electrodes. This problem is particularly pronounced when the recordings are made noninvasively from electrodes placed on the scalp.

Magnetoencephalography (MEG) provides an alternative noninvasive measurement approach with several advantages over scalp and even implanted EEG recordings. Electrical current flow within populations of neurons is a fundamental constituent of brain function. The magnetic fields produced by these electric currents within the brain are measured at the MEG sensor array with high fidelity. Unlike electric fields, magnetic fields do not interact with brain tissue [3,4,5,6]. Therefore, in principle, the currents which are the sources of the measured magnetic field are more readily localized and their time course quantified.

For almost all MEG studies, the neuroelectric sources of the recorded magnetic fields are presumed to be due to population postsynaptic currents within the cerebral cortex [4,5,6]. For this reason, most source level analyses are constrained to identify neuroelectric currents in the cortex only, although there are occasional reports in which neuroelectric dipoles are localized to the white matter—e.g., [7]. Additional remarks are found in the Discussion section.

In the current study, MEG recordings were processed using the referee consensus solver, a method which extracts profuse localized and validated neuroelectric current waveforms (*p* < 10^−12^ for each), not only from cortical and subcortical volumes within the brain, but also from white matter volumes [8,9]. We employed a large set of MEG recordings from healthy volunteers (CamCAN) to generate a normative database, including metrics of test–retest reliability across MEG studies repeated 16 months apart. Good test–retest reliability is a primary requirement for both scientific and diagnostic usefulness. We then analyzed MEG-derived regional neuroelectric measures in a cohort of chronic TBI subjects with persistent symptoms (Targeted Evaluation, Action and Monitoring of Traumatic Brain Injury (Team-TBI)) to identify patterns of brain activity which distinguish between those with and without specific TBI sequelae. Sensitivity to clinical symptoms is a second critical requirement for clinical usefulness.

The primary objective of this work was to obtain and validate clinically useful neuroelectric measures localized within the brain. The *mean* and *standard deviation* for each region from the CamCAN baseline MEG recordings were used to transform the corresponding regional measure for each recording from both cohorts to *z-scores*. The baseline and follow-up CamCAN *z-scores* were used to test for test-retest reliability. The baseline and follow-up Team-TBI *z-scores* were used to test for sensitivity to symptoms. Consistent with the translational objective of the effort, the data processing pipeline was deployed (a) to function without human judgement or intervention and (b) to fully process each new recording within 24 h [8].

The solver identifies high-fidelity waveforms for 80 msec instances of dipole electric current flow localized to about 1 mm^3^ [3,4]. The normalized count of these instances within each standardized region is the measure used in the present study—i.e., each measure is the total value over a brain region automatically identified from one subject’s scan. The fact that each standardized region is automatically identifiable for each scan using Freesurfer [10,11] enables generating standardized measures, i.e., norms, for each region across a large normative cohort. The values for those regions in any individual may then be compared with the norms to assess the normality of the individual’s regional measures.

Note that each measure’s localization is limited by the automated parcellation accuracy of Freesurfer and by the volume of each region. For the CamCAN and TEAM-TBI cohorts reported here, those volumes range from less than 1.0 cm^3^, e.g., left or right nucleus accumbens, to 30–70 cm^3^, e.g., left or right cerebellar cortex.

The results reported here enable rejection of the following null hypotheses. (a) Individuals with and without symptoms are indistinguishable. (b) The cohort membership of each individual (TEAM-TBI or CamCAN) cannot be determined. (c) Regional measures from an individual do not reliably repeat. (d) Within the individual, the neuroelectric activity within a cortical region is indistinguishable from the adjacent rim of white matter.

Joint rejection of hypotheses (a) and (c) supports the potential for these measures as clinically useful in the diagnosis and treatment of insomnia, depression, anxiety, and somatization, common sequelae of TBI. Rejection of hypothesis (a) and (b) suggests the possibility that these measures may be useful as biomarkers for TBI. The emphasis in the study design is towards symptoms rather than etiology. It is hoped that this shift in focus will produce insights which are useful in diagnosis and treatment of those symptoms, regardless of etiology. Repeated MEG recordings during drug or other treatment modalities may provide objective and useful information in assessing treatment efficacy and in making adjustments to the treatment.

## 2. Results

The primary objective of this work was to obtain and validate clinically useful neuroelectric measures localized within the brain. The *mean* and *standard deviation* for each region from the CamCAN baseline MEG recordings were used to transform the corresponding regional measure for each recording from both cohorts to *z-scores*. The baseline and follow-up CamCAN *z-scores* were used to test for test-retest reliability. The baseline and follow-up Team-TBI *z-scores* were used to test for sensitivity to symptoms.

Regional measures of neuroelectric activity for 17 subcortical, 68 cortical, and 18 deep white matter regions were extracted from the MEG recordings for each study participant. For the normative CamCAN cohort, the *mean* and *standard deviation* baseline values for each region are shown in Appendix A. These values were used to transform all regional measures to *z-scores*. The tables also include *correlations* and *differences* for baseline vs. follow-up measures. These were used to assess test-retest reliability. The presence/absence of relationships between these neuroelectric measures and measures of potential clinical relevance was tested.

This effort produced several types of results. Significant relationships were found between measures of tonic regional neuroelectric activity and (1) screening measures of insomnia and psychological distress and (2) subjects in the CamCAN control cohort vs. the TEAM-TBI chronically symptomatic group with history of concussion. These directly impact on the potential usefulness of the measures as diagnostics and as probes for scientific questions.

In addition, (3) both short-term (1 h) and long-term (16-month average) baseline vs. follow-up test-retest reliability results are reported. This too impacts directly on potential clinical utility. Finally, (4) the spatial resolution and statistical power of the neuroelectric measures is demonstrated by identifying significant differential activity between cortical and adjacent white matter regions. 

Note that all regional neuroelectric activity measures were reduced to z-scores; the means and standard deviations of the baseline CamCAN recordings (*n* = 619) were used for the z-score transformation.

### 2.1. Insomnia and Psychological Distress

MEG-derived neuroelectric measures were obtained from 63 TEAM-TBI subjects at baseline and from the 40 who returned for follow-up. Symptom surveys for insomnia and three symptoms of psychological distress were obtained from all but one of the baseline subjects. Standard inventories were for insomnia (Insomnia Severity Index, ISI, [12,13]) and somatization, depression and anxiety (Brief Symptom Inventory, BSI, [14,15]). Cut-offs of 15 (ISI) and 63 (BSI) were used to divide both the baseline and follow-up TEAM-TBI recordings into clinically negative or positive groups.

Regional measures of neuroelectric activity for 17 subcortical and 68 cortical regions were combined into classifiers using stepwise linear classification [16,17,18]. Classification accuracies with *p*-values are shown in Table 1 (Figure 1). The *p*-values were computed as follows. Considering line 1 of said table, 42 of 54 TEAM-TBI subjects who screened negative for insomnia were classified as negative and 12 as positive. The chance that this would happen by chance is equivalent to the odds of getting at least 42 heads when we flip a fair coin 54 times. For each symptom, both sides of the classification have significant *p*-values—i.e., the classifier does well in classifying both those who screen positive and those who screen negative. This provides confidence that the neuroelectric measures which comprise the classifier are related to the symptoms.

The regions whose measures were included are shown in Table 2 (Figure 1). For each symptom, a second classification function was constructed, for which the regions that were included in the first run were excluded. This second run produced significant classification accuracy for insomnia only, as indicated in the tables. This suggests (a) elevated confidence in the relationship between the regions whose neuroelectric measures were used for each classification and (b) that the regional measures included in the second classification function for insomnia are highly correlated with linear combinations of the first set. That is why they were not included in the first classification run.

Classification accuracy reached significance for all four symptoms for both clinically negative and positive groups. These results were weakest for anxiety. Classification accuracy was comparable for baseline and follow-up records; baseline vs. follow-up clinical rating changed for insomnia (*n* = 10), somatization (*n* = 9), depression (*n* = 10), and anxiety (*n* = 15), almost all for the better.

The magnitude of the difference between the nonclinical and clinical groups was examined by testing the difference in the mean scores on each classifier between the groups. This approach provides a complementary view on the magnitude and significance of the differences between the groups achievable via measures of regional neuroelectric activity. The results are shown in Table 3. As might be expected, this produced comparable results to those shown in Table 1, albeit with greater significance.

Finally, to provide information regarding the independence of the results for the different symptoms, coincidence rates for pairs of symptoms and correlations between classifier scores are shown in Table 4. These are the same classifiers reported in Table 1 and the same classifier scores reported in Table 3. In Table 4, the left panel shows that 60% or more of the TEAM-TBI subjects have at least two of the four symptoms. Note that the acceptance criteria for the TEAM-TBI study included “high symptom burden”. The correlations (right panel) are relatively high between pairs of BSI classifier scores and relatively low between the insomnia classifier score and each of the BSI classifiers.

### 2.2. CamCAN vs. TEAM-TBI Cohort

MEG recordings were obtained from 619 CamCAN subjects at baseline, 253 at follow-up, 63 TEAM-TBI subjects at baseline and 40 at follow-up. High resolution T1 weighted MR imaging (MRI) was obtained from all members of both cohorts. Diffusion-weighted imaging (DWI) was obtained from 589 CamCAN subjects, and all of the TEAM-TBI subjects. DWI is required to obtain regional neuroelectric measures for deep white matter tracts.

Regional measures of neuroelectric activity for 17 subcortical and 68 cortical regions were combined into classifiers using stepwise linear classification [16]. Classification accuracies with *p*-values are shown in Table 5. The cortical and subcortical regions which contributed to the classifier in order of their statistical contribution were R thalamus, R cerebellum, R middletemporal, R hippocampus, R lateraloccipital, L isthmuscingulate, L thalamus, L fusiform, brain-stem, and 11 others. When these 20 regions were excluded, classification accuracy remained highly significant albeit reduced. The contributing regions in order were L insula, L lingual, R superior temporal, L caudate, L parahippocampal, and seven others.

As for the cortical and subcortical regions, regional measures of neuroelectric activity for 18 deep white matter tracts were combined into classifiers. Classification accuracies with *p*-values are shown in Table 5. The regions which contributed to the classifier in order were left cortico-spinal tract (L_cst), right inferior longitudinal fasciculus (R_ilf), R_cst, forceps minor (fminor), and six others.

Discriminant analysis reduces all of the measures for an individual to a single score for each classifier. Here, each individual has two scores, one for the cortical/subcortical classifier and one for the deep white matter tract classifier. Figure 2 shows these scores in bivariate plots to illustrate the “spatial” separation which results in the classification accuracies shown in Table 5 (Figure 3).

The components of each classifier are the measures for each included region. The discriminant analysis algorithm identifies those regions for which there are differences between the cohorts and constructs a linear combination of those regions’ measures which adds the differences to enable significant separation/classification.

Figure 4 illustrates the differences for each subcortical region. A dot is plotted for each subject of the CamCAN cohort (upper panel) and the TEAM-TBI cohort (lower panel). Since each measure is a z-score computed from the *mean* and *standard deviation* for that region from the CamCAN cohort, the plots for the CamCAN cohort all have a *mean* of 0.0, *standard deviation* of 1.0, and are nearly symmetrically distributed about the *mean*.

Lack of symmetry primarily results from the fact that each measure has a floor since each measure is a normalized count. The floor for each region is indicated in the figure with a horizontal bar. No individual’s measure for a region can fall below that bar; this results in bunching of the values below the *mean*. This is most pronounced for the accumbens regions for which the floor is only slightly more than one *standard deviation* below the *mean*.

The regions whose contributions to the classifier were greatest are listed above and in Figure 4 (subcortical) and Figure 5 (right cortical). The *means* for each of those regions is markedly displaced from 0.0 as expected.

### 2.3. Test-Retest Reliability

Regional measures of neuroelectric activity for 103 cortical, subcortical, and deep white matter regions were extracted from the resting and task MEG recordings for each subject of both cohorts. The values for the CamCAN cohort were used to assess short-term and long-term test-retest reliability. For short-term reliability, baseline resting vs. task recordings obtained in the same sitting were used. For long-term, baseline vs. follow-up resting recordings were used (*mean* interval = 16 months). The long-term correlations and *mean* differences for each region are listed in Appendix A and are plotted on the y-axis in Figure 6 in blue and red, respectively; the short-term correlations and *mean* differences are plotted on the x-axis. Note that for *mean* difference, the values are z-scores.

The correlations are centered about 0.8 for short-term reliability; for long-term reliability, they are centered around 0.45. The *mean* differences are consistently within 0.03 of 0.0 for the short-term reliability and range between ±0.55 for the long-term reliability. Hence, test-retest reliability is high in the short term and moderate in the long term.

Note that the resting recordings were obtained with eyes closed, while the task recordings were obtained with eyes open. Yet, the comparisons of these, reported here as short-term correlations, are quite high, demonstrating that these measures, unlike fMRI measures, are relatively insensitive to eyes open vs. eyes closed.

### 2.4. Differential Activity: Cortical vs. Adjacent White Matter Regions

For each of 68 cortical regions, Freesurfer identified an adjacent white matter region with a maximum thickness of 5 mm. For each such pair of regions, i.e., cortex and adjacent white matter rim, the difference in activity can be tested for significance by comparing the observed current counts within the regions to the expected counts given the volumes of the regions. This is not only a test of the spatial resolution of the referee consensus solver, but in addition may provide useful neurophysiological information. See the Discussion section for additional comments.

For most regions, there are thousands of counts so there is considerable statistical power to identify differences using the χ^2^ statistic. For each of the 619 baseline CamCAN subjects, there are 68 cortex/white matter region pairs—i.e., 42,092 in total. To reduce false positives due to the large number of comparisons, *p* < 10^−8^ was used as the threshold for significance. In total, 14,187 (33.7%) of the pairs demonstrated greater cortical activity than white matter activity. This supports the claim that the solver’s resolution is less than 5 mm. Surprisingly, 18,129 (43%) of the pairs demonstrated greater white matter activity than cortical activity. Additional comments may be found in the Discussion section.

## 3. Materials and Methods

Magnetoencephalographic (MEG) recordings were processed from each subject of two cohorts: (1) the normative CamCAN cohort, *n* = 619 at baseline, ages 18–87 [19,20], *n* = 253 at follow-up, and (2) the chronically symptomatic concussed TEAM-TBI cohort, *n* = 63 at baseline, ages 21–60, *n* = 40 at follow-up. The MEG recordings were coregistered with high-resolution T1-weighted and DWI weighted MRI scans obtained at baseline.

The primary objective of this work was to obtain and validate clinically useful neuroelectric measures localized within the brain. The *mean* and *standard deviation* for each region from the CamCAN baseline MEG recordings were used to transform the corresponding regional measure for each recording from both cohorts to *z-scores*. The baseline and follow-up CamCAN *z-scores* were used to test for test-retest reliability. The baseline and follow-up Team-TBI *z-scores* were used to test for sensitivity to symptoms. Consistent with the translational objective of the effort, the data processing pipeline was deployed (a) to function without human judgement or intervention and (b) to fully process each new recording within 24 h [8].

The raw MEG data from each subject were initially transformed to a collection of probabilistically validated neuroelectric currents. Each current is 80 msec in duration and is localized in time and space with a resolution of one millisecond (msec) and better than 5 mm (mm). This primary processing step yielded profuse high-resolution neurophysiological measures from within the brain of each subject. The total current count per subject per minute is typically in excess of 500,000.

The current counts were normalized to produce measures of tonic activity for each of 171 standard regions of interest (ROIs): 17 subcortical regions, 68 cortical regions, 68 adjacent white matter regions, and 18 deep white matter tracts. Each regional measure is a count of all the neuroelectric currents localized within the region over the several-minute recording time. The statistical power when comparing counts to test for regional differences is high because the count for each region is high. A final adjustment was applied to each regional measure based on comparable measures obtained from empty-room recordings obtained on the same day.

### 3.1. CamCAN Dataset

The Cambridge Centre for Ageing and Neuroscience (CamCAN) Stage 2 cohort study is a large cross-sectional adult lifespan study (ages 18–87) of the neural underpinnings of successful cognitive ageing [19,20]. The study was conducted in compliance with the Helsinki Declaration, and was approved by the local ethics committee, Cambridgeshire 2 Research Ethics Committee (reference: 10/H0308/50) [20].

The work reported here utilized the majority subset (*n* = 619) of the cohort for whom high-resolution (1 mm) anatomic T1-weighted MR imaging and MEG recordings were available. Of these, 253 follow-up resting recordings were obtained (Stage 3 longitudinal study) with a *mean* interval of 16 months between MEG studies. Diffusion-weighted imaging (DWI) was obtained for 589 of the baseline subjects, 240 of whom returned for follow-up.

MR imaging was obtained on all subjects at a single site using a 3T Siemens TIM Trio scanner with 32-channel head coil. T1 scans were obtained using the MPRAGE sequence. The field of view for these scans was 256 × 240 × 192 at 1 mm resolution. DWI scans were acquired (*n* = 589) with a twice-refocused spin-echo sequence, with 30 diffusion gradient directions for each of the two b-values, 1000 and 2000 s/mm^2^, plus three images acquired with a b-value of 0. Other parameters are: TR = 9100 msec, TE = 104 msec, voxel size = 2.0 mm, FOV = 96 × 96 mm, 66 axial slices [20].

MEG recordings were collected at a single site using a 306-channel VectorView MEG system (Elekta Neuromag, Helsinki). The data were sampled at 1 KHz with antialiasing low-pass filter at 330 Hz and high-pass filter at 0.03 Hz. Continuous head position measures were enabled throughout the recordings. All recordings were obtained with the subject sitting up.

At baseline, 560 s were recorded continuously with eyes closed resting [19]. In the same sitting, 560 s were recorded continuously during performance of a sensorimotor task (*n* = 619). To test for short-term test-retest reliability, MEG-derived measures were compared between the baseline resting and sensorimotor task recordings. The implications of this test are detailed in the Discussion.

For the sensorimotor task, subjects detected visual and auditory stimuli and responded to detection of each with a button press with the right index finger. The stimuli were two circular checkerboards presented simultaneously to the left and right of a central fixation cross, 34 msec duration, and a binaural tone of 300 msec duration. The tone was at 300, 600, or 1200 Hz in equal numbers with the order randomized. In total, 121 trials were presented with simultaneous visual and auditory stimulation. Eight trials were randomly intermixed in which one stimulus was presented at a time—four visual and four auditory. This was carried out to discourage dependence on one stimulus modality only. The average intertrial interval was approximately 4.3 s

At follow-up, 320 s were recorded continuously with eyes closed resting, *n* = 253 [20]. At both baseline and follow-up sessions, 60 s of empty-room recordings were obtained.

### 3.2. TEAM-TBI Dataset

The chronic TBI subject dataset was derived from the Targeted Evaluation, Action and Monitoring of Traumatic Brain Injury (TEAM-TBI) study, a personalized medicine research program for subjects with chronic TBI sequelae at the University of Pittsburgh (clinicaltrials.gov: NCT02657135). All TEAM-TBI subjects gave their informed consent for inclusion before they participated in the study. The study was conducted in accordance with the Declaration of Helsinki, and the protocol was approved by the Institutional Review Board of The University of Pittsburgh (PRO13070121).

Inclusion criteria were ages 18–60 with a history of one or more TBI more than six months prior to the study, with severe persistent chronic TBI sequelae as assessed with the post-concussion symptom severity (PCSS) scale—i.e., high chronic symptom load [21]. In total, 61 of the 63 subjects with MEG recordings had sustained “mild” TBIs. TEAM-TBI subjects underwent a 4-day comprehensive clinical assessment, including advanced neuroimaging, followed by multidisciplinary adjudication of clinical syndromes. TEAM-TBI subjects then completed 6 months of supervised, targeted therapy. Subjects returned to Pittsburgh for a follow-up evaluation (*mean* interval = 6.4 months) to document impact of treatment on identified clinical disorders.

MR imaging was obtained for all subjects at a single site using a 3T Siemens TIM Trio scanner with 32-channel head coil. T1 scans were obtained using the MPRAGE sequence. The field of view for these scans was 256 × 256 × 176 at 1 mm resolution. DWI scans were acquired (*n* = 64) with a twice-refocused spin-echo sequence, with 64 diffusion gradient directions at b-values of 1000 and 3000 s/mm^2^, and 128 directions at b-values of 5000 and 7000. Additional parameters for the four b-values were: TR = 3700, 3700, 4100, 4500 msec, TE = 92, 125, 147, 164 msec. voxel size = 2.4 mm, FOV = 230.4 mm, 63 axial slices.

MEG recordings were collected at a single site using a 306-channel VectorView MEG system, Elekta Neuromag, Helsinki. The data were sampled at 1 KHz with an antialiasing low-pass filter at 330 Hz and high-pass filter at 0.03 Hz. Continuous head position measures were enabled throughout the recordings. All recordings were obtained with the subject sitting up.

At baseline four 200 s resting recordings were obtained with eyes open and fixated with the room darkened (*n* = 63). Four to eight recordings were obtained totaling 1500 s with the lights on during performance of a visual semantic decision task [20]. The protocol was the same at follow-up (*n* = 40).

Baseline resting MEG recordings were used (*n* = 63) for subjects whose high-resolution (1 mm) anatomic T1 and MEG recordings were available. Of these, 40 follow-up resting recordings were obtained—*mean* interval = 6.4 months. DWIs were obtained for 63 of the 64 baseline recordings and 39 of the 40 follow-up recordings. At both baseline and follow-up sessions, 300 s of empty-room recordings were obtained.

### 3.3. MRI Processing

Each high-resolution T1 scan was processed with Freesurfer, version 5.3, using its default Desikan–Killiany atlas parcellation [10,11]. Freesurfer is a segmentation package which automatically and reliably identifies brain regions. The 3-dimensional coordinates of the extent of the brain volume and 153 standardized regions of interest (ROIs) were identified—68 cortical regions, 68 adjacent white matter rims of tissue with thickness ≤ 5.0 mm, and 17 subcortical regions. Figure 7 shows the expected descending relationship between brain volume and age.

Each DWI scan was processed with Tracula, version 1.22 [22]. Tracula is a fully automated package which identifies the 3-dimensional coordinates of the volumes occupied by 18 deep white matter tracts.

### 3.4. MEG Processing

The MEG channels were each filtered using MNE tools with high and low pass at 10 and 250 Hz, 5 Hz roll-off [24]. A value of 250 Hz was used for the low pass to thoroughly remove the continuous head positioning signals present in the raw MEG at 293, 307, 314, and 321 Hz. Previous work has shown that the higher the low-pass frequency, the greater the yield of the solver [8,9]. Note that the 10 Hz high-pass filtering effectively demeans each channel and removes much of the low-frequency content most commonly studied. A value of 10 Hz was used for the high pass because (a) the solver yield significantly increases with the low frequencies removed and (b) the solver was set to search one 80 msec data segment at a time. Data lengths greater than this reduce solver yields [8,9], presumably because current dipole orientation rarely remains stable for that long. This short data length provides very low sensitivity to frequencies below 12 Hz. However, the solver was stepped through the data in 40 msec increments; hence, a bolus of identified current dipoles was identified at 25 Hz. Analysis of the time course of those boluses can provide analysis of low frequencies; however, this is outside the scope of the present study.

For each 1.24 s data segment, mains noise was removed from the CamCAN data at 50, 100, 150, 200, and 250 Hz using polynomial synchronous noise removal [25]. Mains noise was removed at 60, 120, 180, and 240 Hz from the TEAM-TBI data. No other preprocessing was applied and no data segments were excluded by manual artifact identification.

The subject’s head position within the MEG scanner was manually coregistered to the TI scan using Elekta’s Mrilab visualization tool. The coordinates of the center point of a sphere most nearly approximating the brain were identified. These are the only operations in the data processing pipeline for which human judgement was applied. All other operations were fully automated.

Continuous head positioning measures were extracted using Elekta’s MaxFilter tool [26]. The coregistration of the MEG sensor array with the location of the subject’s head and brain was corrected once per second using the continuous head positioning information. This correction was applied to the forward solution used by the solver. The referee consensus solver is described in detail elsewhere. [8,9,27].

The forward solution is the mathematical relationship between a putative electric current within the brain and the resultant magnetic field measurements at the sensor array. The solution we used models the brain as a uniformly conducting sphere [3]. Currents within 30 mm of the center of the sphere are nearly undetectable and the mathematical formulation for the forward solution behaves poorly for this volume; hence, it was excluded from the search. The intersection of this region with an MRI slice is shown in Figure 8. Note that the excluded volume typically includes the posterior thalamus, the posterior commissure, and much of the midbrain (not shown in the figure). The solver’s search volume was delimited using the automated brain segmentation provided by Freesurfer with the 30 mm sphere at the center excluded.

The solver was deployed on The Open Science Grid (OSG), an international distributed supercomputing partnership for data-intensive research [28,29]. The work described here utilized more than 70,000,000 processor-hours on the OSG. The solver is detailed in [8,9] and in Appendix B and Appendix C.

When applied to continuous MEG recordings, the solver typically identifies and validates more than 400 neuroelectric currents within the brain per 40 msec step through the data stream—*p* < 10^−12^ for each and *p* < 10^−4^ for each when conservatively corrected for multiple comparisons (Bonferroni). This is more than 600,000 currents per minute of recorded MEG data identified with millimeter and millisecond resolutions. Note that data segments contaminated by movement or other artifacts were not manually identified for removal. Instead, artifact rejection relied upon the referee consensus solver’s inherent failure to validate neuroelectric currents when presented with noisy data, as shown in Figure 9 [27].

The validated currents within each of the 171 automatically identified brain regions were counted over the duration of the recording. Each count was normalized to current density, ρ_roi_:ρ_roi_ = (count_region_/count_total_) ÷ (vol_region_/vol_total_)(1)

The purpose of this normalization is to enable comparisons of a region within or between individuals, during different states, at different times, or comparisons of one region with another. The normalization is defined so that ρ_region_ = 1.0 for all regions if the neuroelectric currents are uniformly distributed throughout the brain. In the isotropic case, (a) the regional count fraction is always equal to the regional volume fraction and (b) no difference is found for any comparison. Dividing the counts for a region by the total count normalizes ρ for variations due to both data quality and record length. The normalization for data quality is important since the yield of the solver changes from moment to moment as data quality waxes and wanes, e.g., Figure 9 [8,27].

## 4. Normative Measures

Normative values for regional measures of static neuroelectric activity were established. To accomplish this, the *mean*_ρ_ and *standard deviation*_ρ_ for each regional current density (ρ) were obtained from the CamCAN recordings. ρ for any region for any individual may then be compared with the norm for that region by converting it to a *z-score* with a corresponding *p*-value under the assumption that the current densities for the normative population is normally distributed.
*z-score*_ρ_ = (ρ − *mean*_ρ_) ÷ (*sd*_ρ_)(2)

The tables of CamCAN *means* and *standard deviations* are presented in Appendix A. They constitute an atlas which may be used to transform the current densities from any individual to *z-scores* and then assess the normality of deep white matter tonic neuroelectric traffic and cortical/subcortical tonic neuroelectric activity.

Note that transformation of the density measures to *z-scores* nominally equalizes the variances of the norms for all of the regions. This ensures that for a composite measure, e.g., a linear classifier composed of the 18 tract *z-scores*, the impact of each of the 18 densities is approximately equal.

### Empty Room Correction

An ideal method for extracting neuroelectric measures from MEG recordings would consistently yield a value of zero from empty room recordings. The referee consensus solver falls short of this ideal—i.e., there is a significant “dark count”. In addition, the neuroelectric measures extracted from empty room recordings consistently demonstrate significant correlations to measures extracted from human resting recordings obtained on the same day (Figure 10). Each correlation is computed across all of the subjects of one of the cohorts. They are plotted on the y-axis in the figure in blue for the CamCAN cohort (*n* = 619) and in red for the TEAM-TBI cohort (*n* = 63).

Each ρ_region_ was adjusted to compensate for the contribution of the empty-room “dark count”—i.e., the presumed contribution of falsely validated currents. We define the following:

ρ_region-empty_ is the result of the ρ_region_ calculation applied to the empty-room data.

*corr*_region_ is the correlation across the CamCAN subjects between ρ_region_ and ρ_region-empty_—i.e., between the regional activity measured with the subject present and absent in the scanner. Then
ρ_region-corrected_ = ρ_region_ − (*corr*_region_ × ρ_region-empty_)(3)

This correction is an approximate one which uses the correlations as estimates of detecting the competitive advantages that true vs. false neuroelectric currents have. A likely more accurate approach would use a classifier to decide on inclusion or exclusion of one current at a time. For each subject by region, there are typically thousands of identified currents; hence, there is considerable statistical power to accurately train such a classifier. Pursuing this approach is beyond the scope of the work reported here.

## 5. Classification

Regional measures of neuroelectric activity for 17 subcortical and 68 cortical regions were combined into classifiers using stepwise linear classification [16,17,18]. This is an automated computer algorithm which performs discriminant analysis between two groups by computing a linear classification function in a stepwise manner. The groupings for classification were determined by symptom survey scores to test for sensitivity of the measures to symptoms and by cohort membership to test for differences between the cohorts. These are detailed in the Results section.

## 6. Discussion

MEG-derived regional brain measures of tonic neuroelectric activation were tested for long-term test-retest reliability in a large normative cohort, CamCAN, and for sensitivity to symptoms in a chronic TBI cohort, TEAM-TBI. The studied symptoms were insomnia, depression, anxiety, and somatization. Good test-retest reliability was found as well as sensitivity to all four symptoms. Hence, the measures reported here may prove of significant clinical utility in the diagnosis and treatment of these symptoms. In addition, the measures enable classification of each individual into her/his cohort—i.e., normative vs. chronic TBI. Hence, the measures may prove useful as biomarkers for TBI.

The analysis and all results were obtained “by region”. Since we are seeking measures with good test-retest reliability and which can be compared between subjects. The volumetric units we used are regions, i.e., volumes, which can be reliably identified by automated algorithms because they are common to the anatomically normal human brain. As more detailed atlases with finer structures are developed, the measures reported here will be tested for those volumes. For the present, the volumes to which the measures reported here apply are the regions identifiable with Freesurfer 5.3 [10,11] and tracula 1.22.2.12 [22].

Each regional value which demonstrates long-term test-retest reliability is a measure of regional neuroelectric tonus—i.e., the static level of regional neuroelectric activation. Elevated or reduced regional tonus within an individual may prove emblematic of tonic alterations in network function. The ability to assess many such regional measures simultaneously may provide substantive useful information which is complementary to the measures which have specificity to TBI—e.g., bloodborne markers [30,31] and MEG-derived slow waves [32,33,34]. Patterns of altered regional tonus may prove useful in monitoring response to treatment. Analysis of the patterns may enable identification of regions to target for treatment. In particular, the several-centimeter localization of the measures is comparable to the localization precision of transcranial magnetic stimulation (TMS) [35,36,37,38]. The deviations seen in a particular individual may prove sufficient to identify individualized target regions for TMS [39,40] rather than the standardized left and/or right prefrontal cortex currently in use [41,42].

### 6.1. Potential Clinical Utility

This study was undertaken to utilize and assess MEG-derived measures for the diagnosis and monitoring of treatment for chronic symptoms of TBI. We report results which demonstrate (a) sensitivity to the presence/absence of insomnia, somatization, depression, and anxiety (Table 1 and Table 3, Figure 1) and (b) sensitivity to history of concussion and/or chronic symptoms (Table 5, Figure 2, Figure 3, Figure 4 and Figure 5). We cannot directly tie these MEG results to TBI. However, for clinical purposes, the etiology may not matter so long as we can use the measures to more effectively diagnose and treat.

The symptomatic identification accuracies shown in Table 1 and Table 3 are reliably significant, but the percentages are not yet high enough for this classification method to be useful clinically. It is likely that classification accuracy can be increased by (a) refining the measures of neuroelectric activity and by (b) using nonlinear or machine-learning classification methods,

The primary results of the study combine the information contained in many regional neuroelectric measures into patterns of brain activity which are related to chronic symptoms in chronic TBI. We also report cohort-wide differences in regional activity (Figure 4 and Figure 5). These are findings which suggest ways to study the mechanisms which underlie presentation and recovery from symptoms. Productive scientific use of these findings may be complemented by a working theoretical conjecture. To this end, we propose a phantom pain conjecture: all symptoms of psychological distress result from hyperactivity in brain regions responsible for attention and response to pain. In support of this conjecture, many regions which show fMRI-derived differential activation in response to painful stimuli [43] (Table 1) show differential activation in MEG-derived measures in the TEAM-TBI cohort when compared with the CamCAN cohort (Table 6)—e.g., thalamus, precentral, postcentral, and precuneus cortexes, and portions of the cingulate, orbito-frontal, and insular cortexes.

### 6.2. Test-Retest Reliability

We report short-term (1 h, *n* = 619) and long-term (*mean* 16-months, *n* = 253) test-retest reliability for the CamCAN normative cohort for each of the 103 brain regions. The shorter average follow-up for the TEAM-TBI cohort of 6.4 months would be expected to have better test-retest reliability in a normative population. This strengthens confidence in the primary findings of the study that MEG-derived neuroelectric measures change in those TEAM-TBI cohort members whose symptom scores changed.

We used Pearson’s correlation and *mean* difference in test–retest values—Table A1, Table A2 and Table A3 and Figure 6. The difference measure may be used to correct a follow-up measure to compare with baseline.

Short-term test-retest reliability ranged around a *mean* correlation of 0.8 with *mean* average difference well under the *z-score* = 0.03. Long-term test-retest reliability ranged around a *mean* correlation of 0.45, with the *mean* average difference as high as |*z-score*| = 0.55. This is visualized for the hippocampus and supramarginal cortex in Figure 11. These regions were selected because their baseline vs. 16-month follow-up correlations are typical and their difference values are significantly different from zero and visibly so in the figure.

A survey of recent test–retest reliability reports shows reliability ranging widely. As would be expected, test-retest reliability is more consistent for time-locked measures in evoked response paradigms [44,45,46,47,48,49,50,51]. For free-running paradigms, most measures are for resting connectivity confined to frequency bands [52,53,54,55,56,57,58]. The test–retest intervals in most studies are days to weeks. Recasens et al. [47] report results from a 7-week interval and both Piitulainen et al. [57] and Dunkley et al. [48] report results for intervals greater than one year. Over all the reports, the maximum number of subjects was 40 [46]. Cohorts with clinical diagnoses were reported by Candelaria-Cook et al. [55] (psychosis) and Dunkley et al. [48] (PTSD).

Both short-term and long-term reliability values we report compare favorably with all others. For the work reported here (a) the *n* values are much larger, (b) the long-term interval is 16 months, (c) the measures are free-running rather than synchronized to an event, and (d) the measures are from raw rather than averaged data.

### 6.3. Cortical vs. Adjacent White Matter Regions

We report considerable detectable neuroelectric activity from the white matter with positive differentials in favor of adjacent white matter for 56% of those pairs for which the differential is significant with the threshold *p* < 10^−8^. Both previously reported measurements and neurophysiological understanding speak to the plausible validity of these findings. MEG-derived responses from thalamocortical fibers have been reported [59,60]. The source magnetic fields were presumed due to synchronous volleys of action potentials (APs). Each AP produces a travelling current quadrupole. The approximate amplitude was estimated at 100 Amp^−15^ m in an unmyelinated axon [7], with separation of 1 mm between the two dipoles forming the quadrupole, assuming a propagation velocity of 1 m/second. It is presumed that the velocity is greater in the myelinated fibers which comprise the white matter. Hence, the velocity and dipole separation would be greater. This would decrease the distance-dependence of the magnetic field strength and so enhance the detectability of this activity. In addition, the magnetic field, due to an action potential in a single axon, was directly measured at about 150 × 10^−12^ Tesla [60].

A trivial explanation of the profuse findings we report is that cortical activity is localized in nearby white matter due either to poor resolution or to head movements. The robust finding of differential activity between adjacent cortical and white matter ROIs argues against this. As does the design of the method which relies on gradients between points within the brain that are 1 mm apart [8,9] (Appendix B), coupled with the use of one correction per second to the forward solution using continuous head positioning information.

Under the assumption that the white matter is in fact the source of profuse measurable neuroelectric activity, the measured magnetic field components can only be due to synchronous volleys of APs. These would produce transient longitudinal intra-axonal currents which are nearly synchronous in many parallel running axons due to near simultaneous passage of propagating APs.

The detected magnetic field waveforms, e.g., Figure 8, are envelopes which follow the high-frequency waveforms of several AP volleys in sequence. The envelope of a single highly synchronized AP volley would require well under 10 msec to rise and fall. Hence, this type of activity would be dominated by high-frequency contents. This is consistent with the observation that the yield of the solver improves when the low-pass cut-off with which the signals are preprocessed is increased from 150 to 330 Hz [8] (Figure 3). It is also consistent with the observation from a typical task recording that the frequency content of the current waveforms includes profuse resonant activity with frequency content above 70 Hz [61,62].

Additional work outside the scope of the present study is needed to understand the mechanisms which underlie the detection of profuse activity localized to the white matter.

### 6.4. CamCAN vs. TEAM-TBI Differences

The robust differences seen between the two cohorts must be interpreted with caution. We cannot rule out the possibility that these differences are due to differences in the MEG scanners, the scanner noise environments, or for the white matter results to the differences in DWI scan parameters. Given the robustness of the differences, this question can be answered by running a cohort of 40 neurologically normal individuals in the scanner used for the TEAM-TBI cohort. The classifiers developed for the present study can be applied to the measures from such a control cohort. If they are different from the CamCAN cohort, then the differences between the CamCAN and TEAM-TBI cohorts must be presumed to be due to differences in the scanners.

A second potential confounding factor of these results is that the CamCAN resting recordings were obtained with eyes closed, whereas the Team-TBI recordings were obtained with eyes open. To test this, consider that the short test-retest reliability results were obtained by comparing baseline CamCAN resting (eyes closed) with CamCAN sensorimotor task (same sitting, eyes open). The test-retest reliability is very good, i.e., the differences between resting and task are very small. In addition, linear classifiers fail to distinguish between baseline CamCAN resting and task recordings. Hence, this difference in recording conditions, i.e., eyes closed vs. eyes open, does not account for the differences between the cohorts.

It is noteworthy that differences found between TEAM-TBI cohort members with and without specific symptoms is not affected by these questions. The same applies to the test-retest reliability results and to the comparisons between cortical and adjacent white matter volumes. Only the cause of the differences between the cohorts is in question.

Under the assumption that the differences found between cohorts are due to differences from the norm in the neuroelectric brain activity of those with TBI, correspondences between the normal vs. TBI classification results we report and those reported by others may be useful. The high classification accuracy found between CamCAN and TEAM-TBI cohorts, i.e., greater than 90%, provides confidence in the validity of the regions whose measures contribute most to the classifier (Table 6). Of the frontal regions, only the left medial-orbito frontal was a contributor, so significant correspondence was found in our findings to the bilateral orbito-frontal localization of high-amplitude slow waves reported by Lewine et al., Huang et al. and others [32,33,34].

A striking correspondence was also found with a recent study of MEG-derived functional connectivity in TBI [63]. The results of this study highlight the importance of changes in thalamic function in TBI. Table 6 shows that both left and right thalamic measures are major contributors to the classifier and that tonic thalamic activation in the TEAM-TBI cohort members is bilaterally reduced. At present, we can only speculate what the neurophysiologic mechanisms that tie altered regional activation, functional connectivity, and symptoms together are. The growth in our ability to reliably measure such alterations and to target specific regions with drug and TMS therapies may enable us to understand these mechanisms and to more effectively treat them.

## Figures and Tables

**Figure 1 medsci-09-00020-f001:**
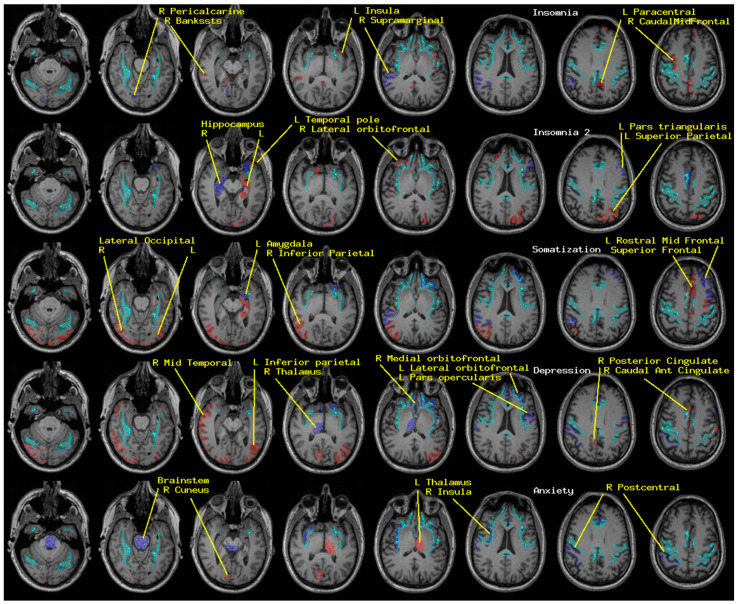
The regions whose neuroelectric activity values contributed to the symptom-specific classifiers reported in Table 1 and Table 2 are shown. Each row shows the regions for the indicated symptom. From top to bottom, they are insomnia, insomnia (2nd step), somatization, depression, and anxiety. The MR imaging (MRI) slices range from left to right, inferior to superior in one-centimeter increments. The left side of each slice is the right brain. Activity in blue/red regions was higher/lower in those who screened positive. The cyan landmarks are the boundaries between gray and white matter in the precentral, cingulate, insula, and fusiform regions.

**Figure 2 medsci-09-00020-f002:**
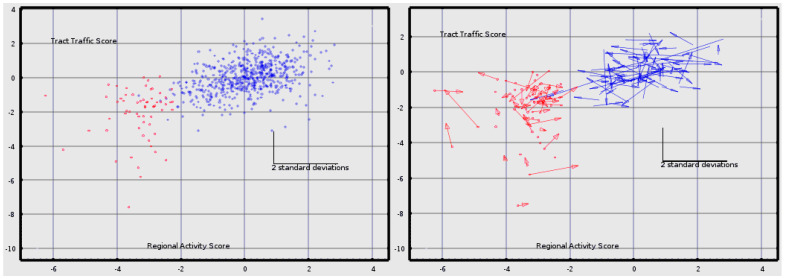
The classifier scores for each baseline CamCAN subject (blue, *n* = 589) and each TEAM-TBI subject (red, *n* = 63) are shown in the **left**. The score on the cortical/subcortical classifier is plotted on the x-axis; the score on the deep white matter classifier is plotted on the y-axis. These are the classifiers whose accuracies are shown in Table 5. In the **right** panel, all baseline TEAM-TBI scores are plotted, *n* = 63. Those who returned for follow-up (*n* = 40) are plotted as arrows; the baseline is plotted at the base of the arrow; the follow-up is plotted at the arrow-head. 63 age and sex matched CamCAN subjects who returned for follow-up are plotted in blue. The standard deviation bars represent 2.0 *standard deviations* for the classifier scores.

**Figure 3 medsci-09-00020-f003:**
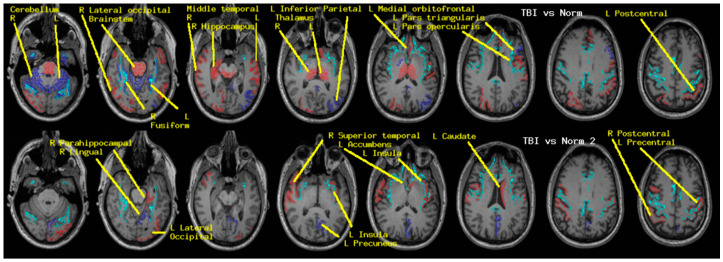
The regions whose neuroelectric activity values contributed to the cohort-specific classifiers reported in Table 5 and Table 6 are shown. The MRI slices range left to right, inferior to superior, in one-centimeter increments. The left side of each slice is the right brain. Activity in blue/red regions was higher/lower in the TEAM-TBI cohort. The cyan landmarks are the boundaries between gray and white matter in the precentral, cingulate, insula, and fusiform regions.

**Figure 4 medsci-09-00020-f004:**
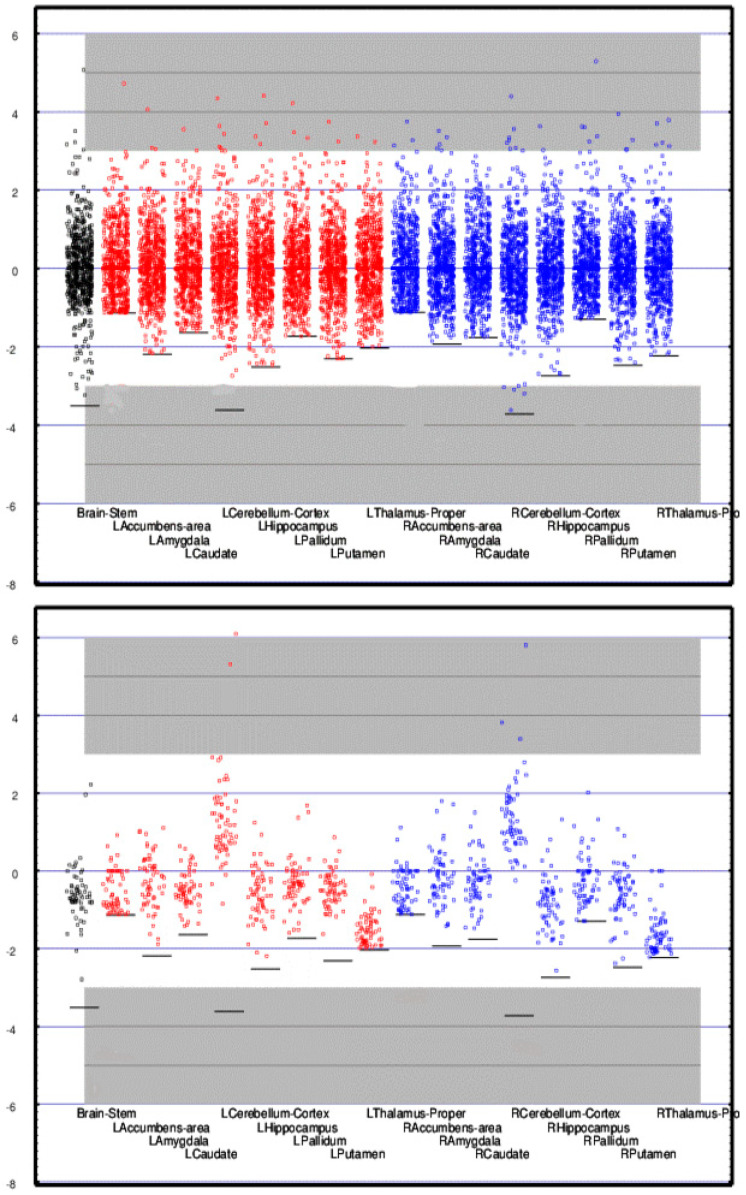
Regional activity (*z-score*) for 17 subcortical regions is shown for each CamCAN (**upper** panel) and TEAM-TBI (**lower** panel) subject. Since the *means* and *standard deviations* for the CamCAN subjects were used to compute the *z-scores*, the *mean* for each of the CamCAN regions is zero and the *z-scores* for each region are distributed approximately normally about zero. The *mean* and Scheme 3. 0 or z > 3.0—i.e., *p <* 0.0014. Individuals whose measures fall into those portions of the graph that are highlighted in gray are far from the norm. Note that assessing sub-*mean* normality is limited by the floor.

**Figure 5 medsci-09-00020-f005:**
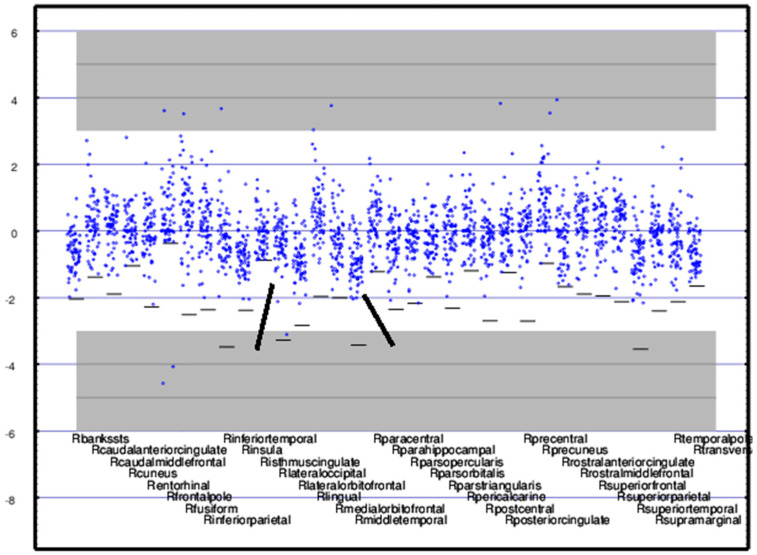
Regional activity (*z-score*) is shown for each TEAM-TBI subject for 34 right cortical regions. As in Figure 2, each horizontal bar represents the floor below which no measure can go. The right middle temporal and lateral occipital cortices, indicated with the wide black bars, contributed significantly to the classifier. Note that the *means* for some regions are greater than zero, some less. See the Figure 4 legend for details.

**Figure 6 medsci-09-00020-f006:**
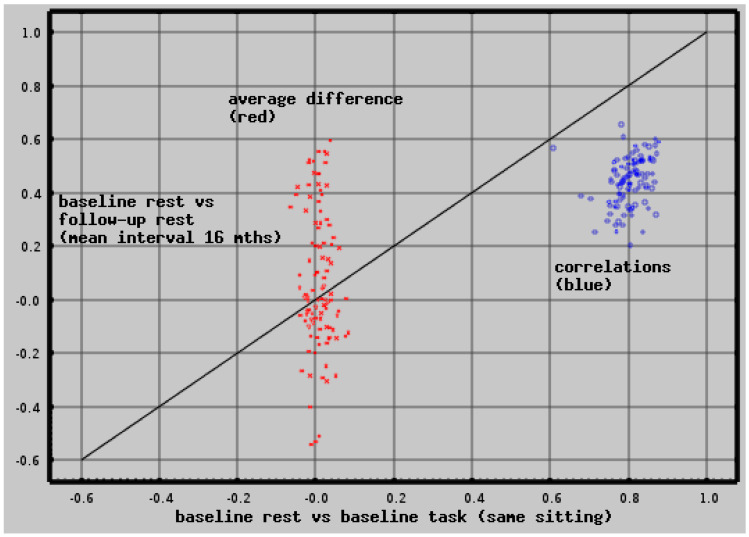
Short-term (same sitting) vs. long-term (*mean* 16-months) test-retest reliability is plotted for each of the 103 cortical, subcortical, and deep white matter regions. Regional measures (*z-scores*) of tonic neuroelectric activity were compared across all CamCAN subjects. Comparisons were between baseline resting and task recordings (short-term reliability, about one hour) and between baseline and follow-up resting recordings (long-term reliability, *mean* 16-month interval). Correlations (blue) and mean differences (red) are plotted—short term (x-axis) vs. long term (y-axis). Long-term test-retest reliability shows reduced correlations and increased differences. Note that mean differences >0.1 or <−0.1 are typically significant with *p* < 0.03.

**Figure 7 medsci-09-00020-f007:**
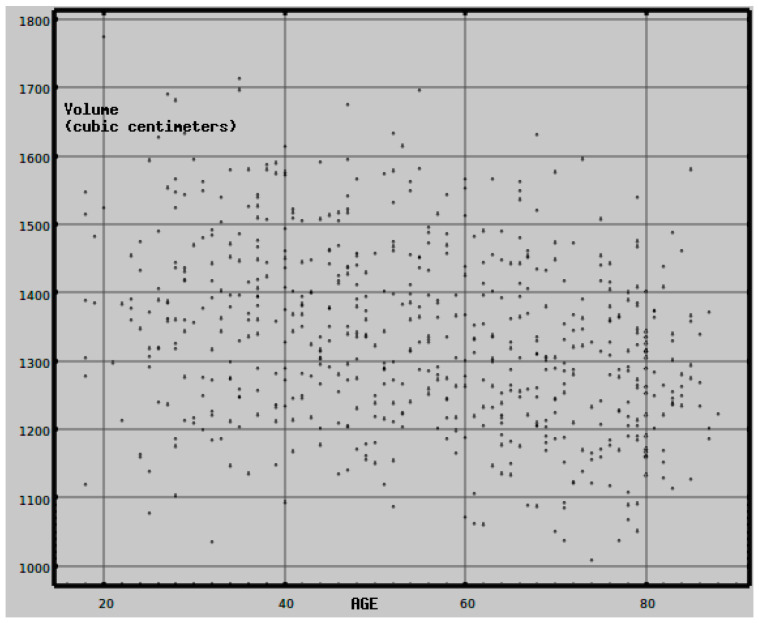
The brain volume for each subject was obtained using Freesurfer (see text). Brain volume for this normative cohort shows the expected decrease with age [23] beginning at about age 60. Over the full age range, the Pearson’s correlation between brain volume and age is −0.295, *df* = 617, *p* < 10^−18^.

**Figure 8 medsci-09-00020-f008:**
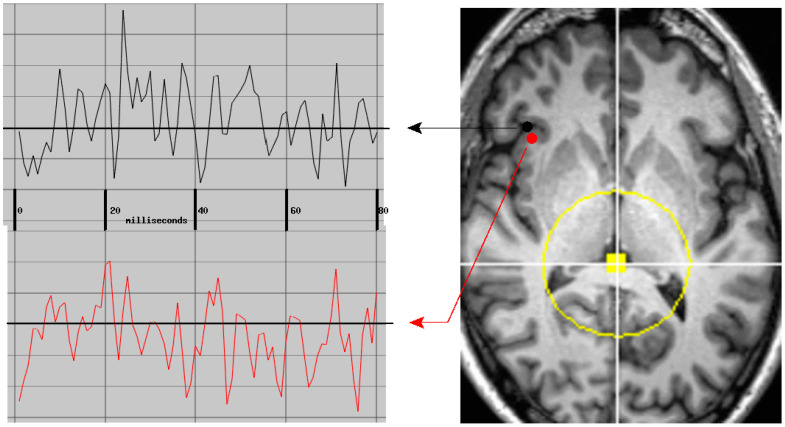
Two typical simultaneously active neuroelectric currents were identified and validated by the referee consensus solver, *p* < 10^−12^ for each—i.e., *p* < 10^−4^ for each when corrected for multiple comparisons. Each waveform has a duration of 80 msec sampled at 1000 Hz. The bandpass is 10–250 Hz. The currents are 5 mm apart with zero-lag cross-correlation of 0.157, *df* = 80, *p* = 0.16. The yellow dot and circle delineate the region near the center of the head which is excluded from the search for neuroelectric currents. See text for details.

**Figure 9 medsci-09-00020-f009:**
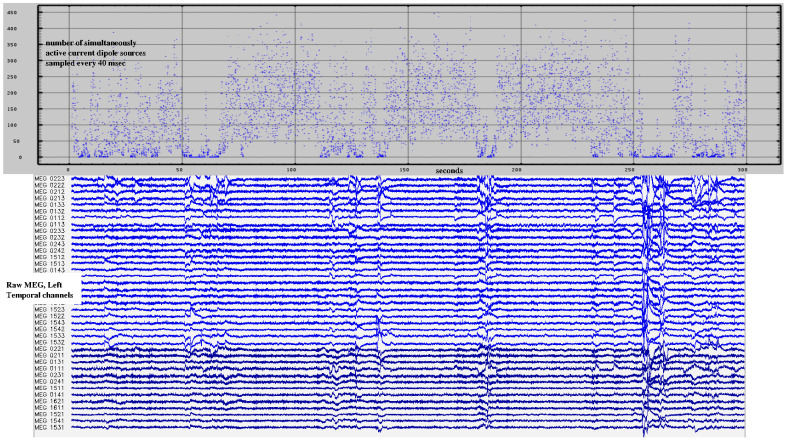
The referee consensus solver automatically fails when the recordings are noisy. 300 s of raw magnetoencephalographic (MEG) (**lower**) and neuroelectric currents (**upper**) are shown. The number of validated (*p* < 10^−12^) currents identified drops markedly when the MEG is noisy.

**Figure 10 medsci-09-00020-f010:**
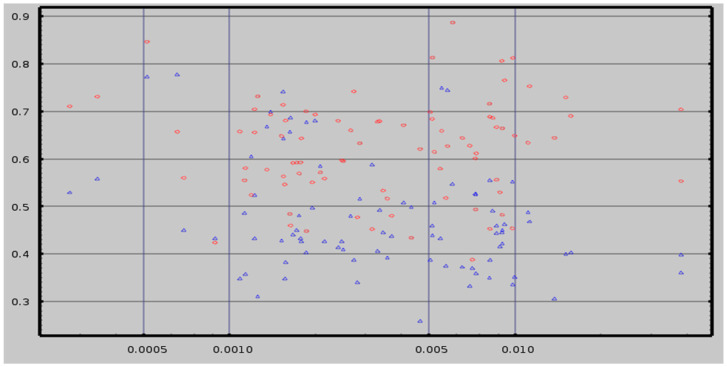
The correlations between measures for empty room and human resting for 85 cortical and subcortical regions for the CamCAN (blue, *n* = 619) and the TEAM-TBI (red, *n* = 63) cohorts are plotted on the y-axis. The number of events found within each region as a fraction of the total number of events found is plotted on the x-axis; the x-axis is logarithmic. The correlations are greater for TEAM-TBI than CamCAN as expected given the large difference in the *n* values. See text for details.

**Figure 11 medsci-09-00020-f011:**
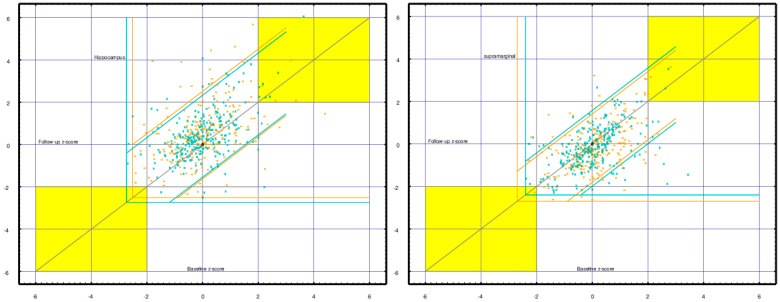
Baseline (x-axis) vs. 16-month follow-up (y-axis) *z-score* regional current densities are plotted for each CamCAN subject (*n* = 253)—right brain in green, left brain in orange. The **left** panel shows the hippocampus. Correlations are 0.456 (L) and 0.476 (R); differences are 0.524 (L) and 0.408 (R). The **right** panel shows the supramarginal cortex. Correlations are 0.524 (L) and 0.482 (R); differences are −0.264 (L) and −0.194 (R). The significance of the differences is readily seen in the displacement of the upward (hippocampus) and downward (supramarginal) distributions. The horizontal and vertical green/orange lines indicate the right/left floor values for the measures. The diagonal lines are 2.0 *z-scores* from the *mean* difference. Yellow highlights the areas for which both baseline and follow-up *z-scores* are ≥2.0.

**Table 1 medsci-09-00020-t001:** Eight-five cortical and subcortical regional measures were trained as linear classifiers by groups determined by symptom survey clinical thresholds. Jackknifed classification accuracies are shown. The results for insomnia (2nd step) are highlighted in gray. These were obtained when the regions which were selected for the first classification analysis were excluded. Symptom survey scores and regional neuroelectric measures were used from both baseline (*n* = 62) and follow-up (*n* = 40) sessions for all subjects of the traumatic brain injury (TEAM-TBI) cohort. See the text for details.

	Classified Negative	Classified Positive	Percentage	*p*-Value
insomnia				
clinically negative	42	12	77.8%	0.000007
clinically positive	11	37	77.1%	0.000031
insomnia—2nd step				
clinically negative	43	11	79.6%	0.0000017
clinically positive	13	35	72.9%	0.00036
somatization				
clinically negative	46	12	79.3%	0.000001
clinically positive	14	30	68.2%	0.0048
depression				
clinically negative	57	13	81.4%	0.00000001
clinically positive	9	23	71.9%	0.0035
anxiety				
clinically negative	37	18	67.3%	0.0032
clinically positive	12	35	74.5%	0.00017

**Table 2 medsci-09-00020-t002:** The regions whose neuroelectric activity values contributed to the symptom-Scheme 1. are shown. Activity in regions marked “+” was higher in those who screened positive; those marked “-” were lower. Regions marked “++” or “--” made the largest statistical contribution to the classifier. The results for insomnia (2nd step) are highlighted in gray. These were obtained when the regions which were selected for the first classification analysis were excluded.

	Insomnia	Somatization	Depression	Anxiety
Brain-Stem				+
L Amygdala		+		
L Hippocampus	--	-		
L Thalamus				--
R Hippocampus	+			
R Thalamus			++	
L caudal middle frontal		-		
L frontal pole	-			
L inferior parietal			--	
L insula	--			
L lateral occipital		-		
L lateral orbito-frontal		+	+	
L paracentral	-			
L pars opercularis			+	
L pars triangularis	+			
L precentral			--	
L rostral anterior cingulate				++
L rostral middle frontal		+		
L superior frontal		-		
L superior parietal	-			
L temporal pole	+			
R bankssts	--			
R caudal anterior cingulate	+	-	-	
R caudal middle frontal	--			
R cuneus			-	-
R inferior parietal		-		
R insula				++
R isthmus cingulate	-			
R lateral occipital		-	--	
R lateral orbito-frontal	-			
R medial orbito-frontal			+	
R middle temporal			-	
R pericalcarine	+			
R posterior central			+	+
R posterior cingulate			-	
R supramarginal	+	+		

**Table 3 medsci-09-00020-t003:** For each symptom, the difference between those who screened negative vs. positive was Table 1. Levene’s *F*-statistic was used to test for significant differences in the variability of the comparison groups. Although no significant differences were found in variability, the most conservative adjusted *df* values were used to compute the *p*-values corresponding to the *t-statistics.* As in Table 1 and Table 2 above, results are highlighted in gray for insomnia (2nd step). These were obtained when the regions which were selected for the first classification analysis were excluded. See the text for details.

	*t*	*df* (Adjusted)	*p*-Value	Levene	*p*-Value
insomnia	9.20	87.8	10^−14^	0.10	0.756
insomnia—2nd step	6.70	86.8	10^−9^	0.39	0.532
somatization	8.63	73.6	10^−13^	1.27	0.262
depression	10.15	56.6	10^−16^	0.09	0.763
anxiety	2.14	90.6	0.025	0.00	0.980

**Table 4 medsci-09-00020-t004:** Left panel: symptom coincidence rate. The fraction of the TEAM-TBI cohort who screened positive for at least two symptoms is shown. For example, 61% of the TEAM-TBI cohort screened positive for both insomnia and depression. Right panel: correlations between classifier scores for the TEAM-TBI cohort. See the text for details.

Symptom Coincidence Rates				Classifier Score Correlations		
Somatization	Depression	Anxiety		Somatization	Depression	Anxiety
0.67	0.61	0.71	insomnia	0.269	0.258	0.383
	0.66	0.70	somatization		0.599	0.666
		0.66	depression			0.781

**Table 5 medsci-09-00020-t005:** 85 cortical and subcortical regional measures were trained as linear classifiers by cohort using the baseline CamCAN and TEAM-TBI measures. Jackknifed classification accuracies are shown for classifiers using 85 cortical and subcortical regions (lines 1–10) and 18 deep white matter tracts (lines 11–15). The results highlighted in gray were obtained when the regions which were selected for the first classification analysis were excluded. See the text for details.

	CamCAN	TEAM-TBI	Percentage	*p*-Value
Cortical/Subcortical				
CamCAN baseline	581	38	93.9%	10^−126^
CamCAN follow-up	227	25	90.1%	10^−42^
TEAM-TBI baseline	0	63	100.0%	10^−18^
TEAM-TBI follow-up	0	40	100.0%	10^−12^
cortical/subcortical—2nd step				
CamCAN baseline	504	115	81.4%	10^−126^
CamCAN follow-up	228	24	90.5%	10^−42^
TEAM-TBI baseline	2	61	96.8%	10^−18^
TEAM-TBI follow-up	3	37	92.5%	10^−12^
deep white matter				
CamCAN baseline	518	71	87.9%	10^−85^
CamCAN follow-up	208	31	87.0%	10^−33^
TEAM-TBI baseline	14	48	77.4%	0.0000024
TEAM-TBI follow-up	9	30	76.9%	0.00015

**Table 6 medsci-09-00020-t006:** The regions whose neuroelectric activity values contributed to the cohort-specific classifiers reported in Table 5 are shown. Activity in regions marked “+” was higher in the Team-TBI cohort; those marked “-” were lower. Regions marked “++” or “--” made the largest statistical contribution to the classifier. The results for the 2nd step are highlighted in gray. These were obtained when the regions which were selected for the first classification analysis were excluded.

Brain-Stem	++	
	Left	Right
Accumbens	-	
Cerebellum	+	++
Caudate	--	
Hippocampus		--
Pallidum		+
Thalamus	--	--
fusiform	++	+
inferior parietal	+	
insula	--	
isthmus cingulate	++	
lateral occipital	-	--
lingual	++	
medial orbito-frontal	-	
middle temporal	-	--
para-hippocampal	--	
pars opercularis	-	-
pars triangularis	+	
peri-calcarine		-
postcentral	-	-
precuneus	+	
precentral	-	
superior parietal		-
superior temporal		--
transverse temporal		-

## Data Availability

The CamCAN datasets are available on request from the Cambridge Centre for Ageing and Neuroscience (CamCAN) at the University of Cambridge, UK, https://www.cam-can.org/, (accessed on 22 March 2021). The TEAM-TBI datasets are available on request from the Federal Interagency Traumatic Brain Injury Research (FITBIR) Informatics System at the National Institutes of Health, USA, https://fitbir.nih.gov/ (accessed on 22 March 2021). The collection of currents identified by the referee consensus solver (≈5 TBytes) are available on request from the corresponding author—kriegerd@upmc.edu.

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
