# Peer review of "Symptom-Dependent Changes in MEG-Derived Neuroelectric Brain Activity in Traumatic Brain Injury Patients with Chronic Symptoms"

_medsci, 2021, doi:10.3390/medsci9020020_

Round 1

Reviewer 1 Report

Difficult to provide a review without directly answering my original critique in the response to reviewers.

Authors are going to have to address my specific comments, thanks

Author Response

Thank you for your helpful comments to our original submission. The requested responses provided a substantive improvement to the quality of the manuscript.

Introduction

The introduction is far too short with limited (barely any) reference to the existing field. The authors talk about the relative strengths of MEG in measuring neurophysiology, especially as it compares to (s)EEG, but there is literally no mention of the background literature preceding this work – extensive work by Huang/Lee, Taylor, Dimitriadis etc are not introduced or even mentioned.

There is no rationale, hypotheses, predictions – why does this study matter? What are the hypotheses the authors are testing? What do they expect to find?

The Introduction has been expanded approximately three-fold. The new material is highlighted in yellow, line numbers 55-61, 72-108. Much of this new material clarifies the motivation for the work, the hypotheses, and what was expected.

The study is focused on identifying measures which may be diagnostics for specific symptoms. It does not attempt to identify the etiology of the symptoms. A key inclusion criterion for the TEAM-TBI cohort was “high symptom burden.” Hence this cohort was a good fit for the questions the study asks. Reference to the work in TBI by Lewine, Huang, and others was added to the opening discussion at lines 595-602 and to the discussion of differences in regional activity between the CamCAN and TEAM-TBI cohorts, highlighted in yellow, lines 729 - 750

Method

What atlas was used to define the ‘standard’ ROIs?

The default Desikan-Killiany atlas was used by freesurfer. This may be found highlighted in yellow at line 440.

Why did the authors filter the low frequency (an often examined marker in mTBI)?

The reasons for the filter pass band are detailed in lines 455-468 and highlighted in yellow.

Were the resting scans eye open or eyes closed? How did the authors deal with eye movement artefacts?

The CamCAN resting recordings were obtained with eyes closed, lines 383and 397, highlighted in yellow. All other recordings for the CamCAN cohort and all recordings for the TEAM-TBI cohort were obtained with eyes open, lines 384-386, 429, Unlike fMRI, little difference was found within an individual between eyes-open and eyes-closed recordings, lines 302-305, 715-722, highlighted in yellow.

How do potential different sites account for the differences seen here? Despite same scanners, are crosssite measures reliable?

This potential confound of the CamCAN vs TEAM-TBI differences and its impact on the current study is treated in detail in the Discussion, lines 706-747, highlighted in yellow.

Results

Rather than listing regions, are the authors able to plot the location of features?

Figures 1 (line 192) and 3 (line 247) contain plots of the regions listed in tables 2 and 6 respectively.

The figures are often difficult to interpret and contrast – for example Fig 2 the groups are stacked above one another; why not side by side to facilitate comparison?

Each panel in figure 4 (was figure 2 in the original manuscript) enables comparison within-cohort of the distribution for one region with another. The panels were placed one above the other to facilitate comparison of the distributions from each region for one cohort with the other. We think it’s more readily interpretable composited this way but it’s no problem to reorganize the figure if the reviewer’s and the editor wish.

Cortical v adjacent white matter regions – does white matter generate an observable signal in MEG? in the final section of results the authors state “This is not only a test of the spatial resolution of the referee consensus solver, but in addition may provide useful neurophysiological information.” – what useful neurophysiological information? The authors then report some differences, with surprising results stating ‘greater white matter than cortical activity’ – what does that mean?

Attempts to localize neuroelectric activity to the white matter are introduced in lines 57-61, highlighted in yellow. These questions are discussed in lines 668-703. That discussion includes informed speculation regards the neurophysiologic genesis of the detected magnetic fields. The present study cannot resolve them but does provide robust evidence that differential activity is readily detected between cortical regions and the adjacent white matter rim.

Discussion

Again, the authors do not discuss their results in light of the extant neurophysiological mTBI literature, or even what this really tells us about the neurophysiology of symptomatic mTBI.

This discussion with references may be found in lines 728-745, highlighted in yellow.

Reviewer 2 Report

The authors have addressed all of my concerns. Therefore I would accept it for publication. However, the results-figures can be presented slightly better eg Fig. 10 stretch and so on. Thus, I will recommend applying these minor changes before publication. 

Author Response

Thank you for your helpful comments to our original submission. The requested responses provided a substantive improvement to the quality of the manuscript.

Reviewer 3 Report

This works by Dr. Okonkwo team is considered novel I n the area of brain trauma in which they utilize human magnetoencephalographic (MEG) recordings as a novel tool to diagnosis therapeutic endpoint to assess chronic sequelae of traumatic brain injury (TBI). applying MEG recordings was performed on a normative cohort (CamCAN and compared to a chronic symptomatic TBI cohort demonstrating good reliability, sensitivity to symptoms (depression nd anxiety...) Comments: the work is considered novel in the are of TBI diagnostics, the only comment om this work is that it was performed on a cohort of mild TBI, it would be of interest if such study can be performed on moderate TBI with or without treatment along with their correlation to biomarker levels such as NFL and other TBI -current assessed inflammatory and TBI-related markers; where the team of Dr Okonkwo is familiar with. Excellent work

Author Response

Thank you for your reading and for the direction you suggest. It is, we think, a likely fruitful one to take in order to identify the TBI-specific processes to which are measures are sensitive.

Reviewer 4 Report

In this study, the authors' test magnetoencephalographic (MEG) derived regional brain measures in the context of chronic traumatic brain injury (TBI). Specifically, this study tests novel MEG-derived regional brain measures of tonic neuroelectric activation for long-term repeat reliability and sensitivity to TBI symptoms. For this study, MEG recordings from a normative cohort (CamCAN baseline = 619, follow-up: n = 258) and a chronic symptomatic TBI cohort (TEAM-TBI, baseline: n = 64, follow-up: n = 39) were compared. The authors claimed that the MEG-derived neuroelectric measures showed good long-term repeat reliability and neuroelectric measures of activation. Such parameters were sensitive to the presence/absence of these symptoms in the TBI group. Moreover, the authors claim that the novel regional MEG-derived neuroelectric measures (obtained and tested in this study) can be clinically useful. Overall, the study seems relevant to the research topic. Imaging techniques and MEG were performed using standard corrections and protocols. Inclusion and exclusion criteria are listed. The sample size is significant, the statistical analysis seems appropriate, and the results well presented addressing the main research questions. The discussion is constructive leading to logical conclusions. However, some minor concerns and remarks need to be addressed before this interesting work can move forward.

1. The authors mentioned that the primary objective of this work was to obtain and validate clinically useful neuroelectric measures localized within the brain. For the readers not familiar with this technology, has been MEG previously compared / Validated with the current clinical standard of care methods such as EEG & deep brain recording? If so, the addition of few representative references could be helpful.

2. The study of MEG in the context of TBI has been extensively studied in the last decade (Vakorin, VA et al 2016; PLOS Computational Biology & Antonakakis et al 2016; International Journal of Physiology & Huang, M-X et al 2020; Journal of Neurotrauma, among many others). Might be the author can briefly stress what are the novel & practical insights that this particular study offers to the scientific & clinical TBI community.

3. Please clarify if the experiment conducted in the CamCAN group was also approved by institutional review board protocols, as described by the TEAM-TBI cohort (PRO13070121). This is a crucial statement.

4. Regarding the cohorts, demographics variables (age, gender, etc), details of the TBI characteristics (mild, moderate, severe), mechanism of injury (automobile accidents, falls, sports injuries), associated medical or surgical treatments, (among others). were not revealed. Was this information not available from the clinical records? Optionally, maybe a small table can help to visualize the characteristics of both groups, and potentially guide the authors to extract more clinically relevant information on their future studies.

5. The authors mentioned that baseline and follow-up CamCAN z-scores were used to test for repeat reliability. On the other hand, the baseline and follow-up of Team-TBI z-scores were used to test for sensitivity to symptoms. Nonetheless, the follow-up time points for the control (CamCAN, 16- 17-month follow-up) and TBI groups (TEAM-TBI, 6-month follow-up) are indeed different. Could such mismatched timelines able to potentially affect the overall comparative results? A brief explanation of the rationale will suffice.

6. Why the anatomical (T1) and DWI sequences & scanning protocols are substantially different between the CamCan and the TEAM-TBI datasets? In the case of DWI white matter (WM) reconstruction, could such differences in the number of diffusion gradient directions and b-values able to affect the fittings and subsequently the WM topography? A brief justification could help to clarify this point.

7. In terms of exclusion criteria for the TEAM-TBI dataset, did the author's considered if some of the patient population was potentially under post-TBI medications that could potentially affect the electric brain activity (barbiturates, benzodiazepine, antiepileptics, etc)?

Minor remarks

8. On page 7, line 235; for the sentence “… the score on the deep white matter classiFigure 5…”, please revise.

9. On page 14, line 430; “…and All baseline resting MEG recording..”, please remove the word "all" if necessary.

10. On page 15, line 480; for the edict “...solver is described in detail elsewhere.[8,9,27]”, please correct the end of the sentence punctuation.

11. On page 19, lines 590-591; for the statement “…with freesurfer 5.3 [10,11] and tracula 1.22.2.12..”, the authors may consider properly include the version of the mentioned software in the material and methods section.

In sum, the current study represents a valid attempt to use parameters derived from MEG, as a useful technique to detect clinical biomarkers in the context of chronic TBI.

Author Response

Thank you for your helpful comments. The requested responses provided a substantive improvement to the quality of the manuscript.

  1. The authors mentioned that the primary objective of this work was to obtain and validate clinically useful neuroelectric measures localized within the brain. For the readers not familiar with this technology, has been MEG previously compared / Validated with the current clinical standard of care methods such as EEG & deep brain recording? If so, the addition of few representative references could be helpful.
  2. The study of MEG in the context of TBI has been extensively studied in the last decade (Vakorin, VA et al 2016; PLOS Computational Biology & Antonakakis et al 2016; International Journal of Physiology & Huang, M-X et al 2020; Journal of Neurotrauma, among many others). Might be the author can briefly stress what are the novel & practical insights that this particular study offers to the scientific & clinical TBI community.

Reference to the work in TBI by Lewine, Huang, and others is found in the opening to the discussion at lines 595-602 and in the discussion of of differences in regional activity between the CamCAN and TEAM-TBI cohorts, highlighted in yellow, lines 729 – 750.

  1. Please clarify if the experiment conducted in the CamCAN group was also approved by institutional review board protocols, as described by the TEAM-TBI cohort (PRO13070121). This is a crucial statement.

Institutional Ethics Committee approval (reference; 10/H0308/50) is now included in the text at lines 362-364 (highlighted in cyan)

  1. Regarding the cohorts, demographics variables (age, gender, etc), details of the TBI characteristics (mild, moderate, severe), mechanism of injury (automobile accidents, falls, sports injuries), associated medical or surgical treatments, (among others). were not revealed. Was this information not available from the clinical records? Optionally, maybe a small table can help to visualize the characteristics of both groups, and potentially guide the authors to extract more clinically relevant information on their future studies.

The inclusion criteria for the TEAM-TBI study are summarized in lines 408-411. 61 of the 63 for which baseline MEG studies were obtained had had a mild TBI by their report to us, i.e. no attempt was made to obtain clinical records. This is one of the reasons that the study focused on symptoms rather than etiology.

  1. The authors mentioned that baseline and follow-up CamCAN z-scores were used to test for repeat reliability. On the other hand, the baseline and follow-up of Team-TBI z-scores were used to test for sensitivity to symptoms. Nonetheless, the follow-up time points for the control (CamCAN, 16- 17-month follow-up) and TBI groups (TEAM-TBI, 6-month follow-up) are indeed different. Could such mismatched timelines able to potentially affect the overall comparative results? A brief explanation of the rationale will suffice.

A statement about this has been added to the discussion at lines 638-641.

  1. Why the anatomical (T1) and DWI sequences & scanning protocols are substantially different between the CamCan and the TEAM-TBI datasets? In the case of DWI white matter (WM) reconstruction, could such differences in the number of diffusion gradient directions and b-values able to affect the fittings and subsequently the WM topography? A brief justification could help to clarify this point.

A statement about this has been added to the discussion at lines 713-714

  1. In terms of exclusion criteria for the TEAM-TBI dataset, did the author's considered if some of the patient population was potentially under post-TBI medications that could potentially affect the electric brain activity (barbiturates, benzodiazepine, antiepileptics, etc)?

Many of our subjects reported that they were using medications, alcohol, and other drugs. This is a problem in the chronically symptomatic TBI population, particularly in combat veterans who comprised two thirds of the cohort. We would have had great difficulty filling the study had drug and/or medication use been an exclusion criterion.

Minor remarks

  1. On page 7, line 235; for the sentence “… the score on the deep white matter classiFigure 5…”, please revise.

This error is an artifact of formatting changes applied by the MDSI to create the reviewers’ copy. I’ve check to make sure it’s correct in the primary document.

  1. On page 14, line 430; “…and All baseline resting MEG recording..”, please remove the word "all" if necessary.

This is no corrected.

  1. On page 15, line 480; for the edict “...solver is described in detail elsewhere.[8,9,27]”, please correct the end of the sentence punctuation.

This error is an artifact of formatting changes applied by the MDSI to create the reviewers’ copy. I’ve check to make sure it’s correct in the primary document.

  1. On page 19, lines 590-591; for the statement “…with freesurfer 5.3 [10,11] and tracula 1.22.2.12..”, the authors may consider properly include the version of the mentioned software in the material and methods section.

I’ve made sure that the version numbers are included in Methods as you suggest. They are at lines 439 and 446.

In sum, the current study represents a valid attempt to use parameters derived from MEG, as a useful technique to detect clinical biomarkers in the context of chronic TBI.

This manuscript is a resubmission of an earlier submission. The following is a list of the peer review reports and author responses from that submission.

Round 1

Reviewer 1 Report

In this manuscript Krieger et al investigate neuroelectric activity in TBI, compared to a normative control sample. Patterns of MEG activity can distinguish symptomatic participants.

While the study is timely and important, and powerful in terms of the sample size and approach, numerous issues dampen my enthusiasm for the paper. I would currently recommend rejection of the study. What follows is a section by section critique of the paper.

Introduction

The introduction is far to short with limited (barely any) reference to the existing field. The authors talk about the relative strengths of MEG in measuring neurophysiology, especially as it compares to (s)EEG, but there is literally no mention of the background literature preceding this work – extensive work by Huang/Lee, Taylor, Dimitriadis etc are not introduced or even mentioned.

There is no rationale, hypotheses, predictions – why does this study matter? What are the hypotheses the authors are testing? What do they expect to find?

Method

What atlas was used to define the ‘standard’ ROIs?

Why did the authors filter the low frequency (an often examined marker in mTBI)?

Were the resting scans eye open or eyes closed? How did the authors deal with eye movement artefacts?

How do potential different sites account for the differences seen here? Despite same scanners, are crosssite measures reliable?

Results

Rather than listing regions, are the authors able to plot the location of features?

The figures are often difficult to interpret and contrast – for example Fig 2 the groups are stacked above one another; why not side by side to facilitate comparison?

Cortical v adjacent white matter regions – does white matter generate an observable signal in MEG? in the final section of results the authors state “This is not only a test of the spatial resolution of the referee consensus solver, but in addition may provide useful neurophysiological information.” – what useful neurophysiological information? The authors then report some differences, with surprising results stating ‘greater white matter than cortical activity’ – what does that mean?

Discussion

Again, the authors do not discuss their results in light of the extant neurophysiological mTBI literature, or even what this really tells us about the neurophysiology of symptomatic mTBI

Reviewer 2 Report

The authors of the present study tried to exploit a big dataset of MEG-MRI recordings from patients with traumatic brain injuries with the main goal, the diagnosis, and treatment.

The paper contains a lot of useful results however is not well structured and the study contains many limitations.

For example, the abstract is too long (many details) and not formally written (keep to 250 words). The number of keywords is large (keep them to less than 5). Please elaborate.

The word ‘neuroelectric’ has no meaning for the description of the MEG recordings. Do the authors mean neurophysiological MEG recordings? Please elaborate accordingly.

Introduction, paragraphs 1, 3, and 4. The authors need to add references to supports their arguments.

Introduction, paragraph 2: It is not clear whether epileptologists do this task or other types of neurologists. Please elaborate on the specific sentence towards the topic of the present manuscript.

Section “Insomnia and psychological distress” What do the authors mean with 'low p-values.' The characterization does not fit. A p-value is used to declare statistical significance. Please elaborate.

Section “CamCAN dataset”, paragraph 3, A limitation in this study is that the subjects have measured in sitting position. Based on Rice et al., 2013

(Rice, J. K., Rorden, C., Little, J. S., & Parra, L. C. (2012). Subject position affects EEG magnitudes. NeuroImage, 64, 476–484. https://doi.org/10.1016/j.neuroimage.2012.09.041),

possible inaccuracies can appear during the source reconstruction due to the existence of CSF between brain and skull tissues

(See Antonakakis M, Schrader S, Wollbrink A, Oostenveld R, Rampp S, Haueisen J, Wolters CH. The effect of stimulation type, head modeling, and combined EEG and MEG on the source reconstruction of the somatosensory P20/N20 component. Hum Brain Mapp. 2019 Dec 1;40(17):5011-5028. doi: 10.1002/hbm.24754)

for a comparison between source reconstructions with the head model with CSF and without (Fig 4).

Please add this limitation in the present manuscript, citing the last study that contains the Rice et al., 2013, too.

Section “MRI processing”: Indeed, with the freesurfer toolbox, we can segment automatically brain tissue, however, there are might be errors in the segmented result (segmented brain). To avoid re-run the entire study the reviewer suggests discussing this limitation in their discussion.

Section “MEG processing”, paragraph 1: The authors have not taken into account ocular or cardiac artifacts. There are automatic procedures that can take care of the ocular and cardiac artifacts (see Antonakakis M, Dimitriadis SI, Zervakis M, Micheloyannis S, Rezaie R, Babajani-Feremi A, Zouridakis G, Papanicolaou AC. Altered cross-frequency coupling in resting-state MEG after mild traumatic brain injury. Int J Psychophysiol. 2016 Apr;102:1-11. doi: 10.1016/j.ijpsycho.2016.02.002 . Epub 2016 Feb 22. PMID: 26910049).

Please refer to the specific limitation of this manuscript and add corresponding references (e.g. Antonakakis et al., 2016)

In the same section, paragraph 4, end of the section: The authors used a simplistic head model, neglecting the effect of the CSF or the white matter anisotropy, even though there is a large survey that shows such effects. The most recent studies are Antonakakis et al., 2019 HBM, and Schrader et al., 2020. Please discuss and cite this limitation with the above references.

Antonakakis M, Schrader S, Wollbrink A, Oostenveld R, Rampp S, Haueisen J, Wolters CH. The effect of stimulation type, head modeling, and combined EEG and MEG on the source reconstruction of the somatosensory P20/N20 component. Hum Brain Mapp. 2019 Dec 1;40(17):5011-5028. doi: 10.1002/hbm.24754

Schrader S, Antonakakis M, Rampp S, Engwer C and Wolters CH. A novel method for calibrating head models to account for variability in conductivity and its evaluation in a sphere model. https://iopscience.iop.org/article/10.1088/1361-6560/abc5aa

Section “Discussion”, end of paragraph 1, the authors could add and compare their results with other studies relative to TBI or mTBI diagnosis using classification approaches.

Recent examples are Antonakakis et al., 2020, Vergara et al., 2018 and

Antonakakis M, Dimitriadis SI, Zervakis M, Papanicolaou AC, Zouridakis G. Aberrant Whole-Brain Transitions and Dynamics of Spontaneous Network Microstates in Mild Traumatic Brain Injury. Front Comput Neurosci. 2020;13:90. Published 2020 Jan 15. doi:10.3389/fncom.2019.00090 

Vergara, V. M., Mayer, A. R., Kiehl, K. A., and Calhoun, V. D. (2018). Dynamic functional network connectivity discriminates mild traumatic brain injury through machine learning. Neuroimage Clin. 19, 30–37.doi: 10.1016/j.nicl.2018.03.017

Vakorin, V. A., Doesburg, S. M., da Costa, L., Jetly, R., Pang, E. W., and Taylor, M. J. (2016). Detecting mild traumatic brain injury using resting state magnetoencephalographic connectivity. PLoS Comput. Biol. 12:e1004914. doi: 10.1371/journal.pcbi.1004914

Section „ Cortical vs adjacent white matter regions “, it is not clear whether the DTI directions were enough in order to reconstruct the Fractional Anisotropy due to the possibility of crossing fiber. Please elaborate and add the corresponding limitation in the limitation section of the manuscript.

In general, the entire manuscript and specifically the discussion seems to be written in hurry. In addition, the discussion is quite limited. There is no limitation section, nor conclusion. The authors need to restructure their work in order to include a complete structure with also more informative, complete details in their figures and better representation of the images.

Finally yet importantly, the authors always mention for classification or that the classifier showed something. However, there is no references or description (in the methods sections) about classification. What do the authors mean? Is it a classification in terms of machine learning? Alternatively, manual human categorization of the metadata? Please provide necessary details of information about the classification.